# Risk Factors for Influenza-Induced Exacerbations and Mortality in Non-Cystic Fibrosis Bronchiectasis

**DOI:** 10.3390/v15020537

**Published:** 2023-02-14

**Authors:** Hung-Yu Huang, Chun-Yu Lo, Fu-Tsai Chung, Yu-Tung Huang, Po-Chuan Ko, Chang-Wei Lin, Yu-Chen Huang, Kian Fan Chung, Chun-Hua Wang

**Affiliations:** 1Department of Thoracic Medicine, Chang Gung Memorial Hospital, Taipei 105, Taiwan; 2College of Medicine, Chang Gung University, Taoyuan 333, Taiwan; 3Department of Thoracic Medicine, New Taipei City Municipal TuCheng Hospital, Chang Gung Medical Foundation, New Taipei City 236, Taiwan; 4Center for Big Data Analytics and Statistics, Chang Gung Memorial Hospital, Taoyuan 333, Taiwan; 5Airway Disease Section, National Heart and Lung Institute, Imperial College London, London W2 1PG, UK; 6Biomedical Research Unit, Royal Brompton Hospital, London SW7 2BX, UK

**Keywords:** bronchiectasis, influenza, bacteria, acute exacerbations, co-infection

## Abstract

Influenza infection is a cause of exacerbations in patients with chronic pulmonary diseases. The aim of this study was to investigate the clinical outcomes and identify risk factors associated with hospitalization and mortality following influenza infection in adult patients with bronchiectasis. Using the Chang Gung Research Database, we identified patients with bronchiectasis and influenza-related infection (ICD-9-CM 487 and anti-viral medicine) between 2008 and 2017. The main outcomes were influenza-related hospitalization and in-hospital mortality rate. Eight hundred sixty-five patients with bronchiectasis and influenza infection were identified. Five hundred thirty-six (62%) patients with bronchiectasis were hospitalized for influenza-related infection and 118 (22%) patients had respiratory failure. Compared to the group only seen in clinic, the hospitalization group was older, with more male patients, a lower FEV_1,_ higher bronchiectasis aetiology comorbidity index (BACI), and more acute exacerbations in the previous year. Co-infections were evident in 55.6% of hospitalized patients, mainly caused by *Pseudomonas aeruginosa* (15%), fungus (7%), and *Klebsiella pneumoniae* (6%). The respiratory failure group developed acute kidney injury (36% vs. 16%; *p* < 0.001), and shock (47% vs. 6%; *p* < 0.001) more often than influenza patients without respiratory failure. The overall mortality rate was 10.8% and the respiratory failure group exhibited significantly higher in-hospital mortality rates (27.1% vs. 6.2%; *p* < 0.001). Age, BACI, and previous exacerbations were independently associated with influenza-related hospitalization. Age, presence of shock, and low platelet counts were associated with increased hospital mortality. Influenza virus caused severe exacerbation in bronchiectasis, especially in those who were older and who had high BACI scores and previous exacerbations. A high risk of respiratory failure and mortality were observed in influenza-related hospitalization in bronchiectasis. We highlight the importance of preventing or treating influenza infection in bronchiectasis.

## 1. Introduction

Bronchiectasis is characterized by abnormal dilatation of the bronchi and systemic inflammation, and such patients are at an increased risk for respiratory tract infections [1,2], which play a major role in causing exacerbations of bronchiectasis. These exacerbations are associated with more severe disease and a higher mortality rate [3]. While the main pathogen for exacerbations is bacteria [3,4], other pathogens such as fungi, nontuberculous mycobacteria (NTMs), and viruses have been implicated but remain less well documented [5]. Viral infections such as rhinovirus, coronavirus, and influenza viruses may trigger exacerbations of bronchiectasis [6,7] and the virus is more frequently detected during the exacerbation than during a period of stability [6,7,8]. There is a paucity of clinical studies that have specifically investigated the role of influenza infection in clinical outcomes of bronchiectasis.

The interactions between pathogenic bacteria, viruses, and host defense responses, such as the immune reaction, epithelial injury, and repair delay after virus infection, have been demonstrated in chronic respiratory diseases, and may frequently lead to a lethal synergism in susceptible people with more severe disease [9,10]. In critical patients infected with influenza, bacterial co-infection and its complications, such as development of acute respiratory distress and acute kidney injury, contribute to the mortality associated with influenza [11,12,13,14,15]. In our recent study, the common existing bacterial pathogens in acute exacerbations of bronchiectasis were *Pseudomonas aeruginosa*, followed by *Klebsiella pneumoniae* and *Haemophilus influenzae* [16]. Whether this underlying chronic infection leaves bronchiectasis patients particularly vulnerable to influenza infections is not known. There are very limited studies that report the clinical outcomes and risk factors of viral infection during bronchiectasis exacerbations. The aim of this study was therefore to survey the clinical manifestations and prognoses of influenza infection in patients with bronchiectasis and to investigate the risk factors associated with influenza-related hospitalization and in-hospital mortality.

## 2. Methods

### 2.1. Bronchiectasis Cohort

This study analyzed the data of a multi-institutional bronchiectasis cohort in Chang Gung Research Database (CGRD). The CGRD provides the electronic medical records collected from the Chang Gung Memorial Hospital system, which includes three medical centers and four regional hospitals, as reported previously [17,18]. Patients with at least two bronchiectasis diagnoses (International Classification of Diseases, 9th Clinical Modification (ICD-9-CM) 494.0 or 494.1) from outpatient visits or from hospitalization records were identified in the cohort [19]. The diagnosis of bronchiectasis was made from clinical symptoms, history, and a high-resolution CT (HRCT) of the lungs by a radiologist and pulmonary specialist. The Institutional Review Board of Chang Gung Memorial Hospital approved this study (IRB number: 201800712B0C502).

### 2.2. Inclusion Criteria

This cohort included adult patients (aged ≥ 18 years) with diagnoses of bronchiectasis recorded in CGRD between January 2006 and June 2016. The inclusion criteria were bronchiectasis patients who had typical influenza-like illnesses with diagnoses of influenza (ICD-9-CM code 487) and concomitant use of anti-virus medicines [20,21,22,23]. Clinicians made the diagnoses of influenza infection based on typical influenza-like symptoms, positive reverse-transcription polymerase chain reactions (PCR), or influenza rapid antigen tests. The definition of influenza infection has been validated in previous publications of the national health insurance database of Taiwan [20,21,22,23].

### 2.3. Main Outcomes

The primary outcome was influenza-related severe exacerbations, defined as hospitalization for diagnoses of ICD-9-CM code 480–487 [20,24]. The secondary outcome was in-hospital mortality. Acute respiratory failure was defined by ICD-9-CM code 518.81 or 518.82, or ICD-10-CM code J96.0 with mechanical ventilator use [25].

### 2.4. Clinical Parameters

We retrieved demographic data, CT images, laboratory, microbiology and pulmonary function reports. The bronchiectasis aetiology comorbidity index (BACI) was calculated for each subject based on comorbidities obtained from CGRD diagnoses (ICD-9-CM and ICD-10) [26]. The etiology of bronchiectasis was determined as described in a previous study [19]. Clinicians requested an immune or autoimmune screen if signs of immunodeficiency and connective tissue diseases were present. When primary ciliary dyskinesia was suspected, nasal mucociliary clearance was measured by using the saccharin test. Evaluations of α1-antitrypsin were performed when an HRCT demonstrated the presence of emphysema affecting the lower lobes. Sweat tests were requested if signs and symptoms suggestive of cystic fibrosis were present. We collected sputum microbiology reports during hospitalization. Pulmonary bacterial co-infection was defined as the presence of pneumonia and a positive bacterial culture in sputum or bronchoalveolar lavage fluid during the period of hospitalization [12]. Shock was defined as the necessity for use of parenteral inotropic agents or vasopressors. A pulmonary function test was performed with a spirometer according to the American Thoracic Society and the European Respiratory Society criteria [27]. Medical treatment included anti-viral medicine, antibiotics, systemic corticosteroids, and inhalation medication. Acute kidney injury was defined as an increase in serum creatinine level of 50% or 0.3 mg/dL above baseline during hospital admission [28].

### 2.5. Statistical Analysis

Chi-square tests and two-sided Fisher exact tests were used for dichotomous variables, unpaired *t*-tests for normally distributed continuous variables, and Mann-Whitney U tests for non-normally distributed continuous data. *p*-values (two-sided) < 0.05 were considered statistically significant. A univariate descriptive analysis was performed to identify risk factors for hospitalization and mortality in patients with bronchiectasis and influenza infection. Variables with a significance level of *p* < 0.05 were selected. Next, a multivariate Cox proportional hazards regression was used to identify independent risk factors. An ROC analysis was performed to validate the Cox model. Statistical analyses were performed using SAS software, version 9.4 (SAS Institute, Cary, NC, USA).

## 3. Results

There were 9516 patients with bronchiectasis recovered from CGRD between 2008 and 2017. A total of 865 bronchiectasis patients having influenza infection were identified. Figure 1 showed the incident number in each year. Five hundred thirty-six (62%) patients with bronchiectasis were hospitalized for influenza-related infection. The demographic and clinical characteristics are summarized in Table 1. Compared to the clinic group (defined as no episode of hospitalization), the hospitalization group was older, with more male patients, lower FEV_1_, higher BACI index, and higher acute exacerbation rates in the previous year. The hospitalization group had higher proportions of pre-existing COPD, connective tissue disease, and diabetes mellitus than the clinic group. The usage rate of anti-viral agents was similar in both groups. A higher proportion of bronchiectasis patients in the hospitalization group received antibiotics (94.9% vs. 9.1%, *p* < 0.001) and systemic corticosteroids (54.6% vs. 11.3%, *p* < 0.001) than those in the clinic group. Figure 2 showed the annual incident numbers of bronchiectasis with influenza infection in CGRD since 2008–2017.

The characteristics of the hospitalized patients with bronchiectasis and influenza infection are demonstrated in Table 2. 118 (22%) patients had respiratory failure, with a mean age of 70.5 years. Patients with or without respiratory failure had similar ages, as well as lung function, BACI index, and acute exacerbations rates in the previous year. Bronchiectasis patients with respiratory failure exhibited significantly higher levels of white blood cell count as well as C-reactive protein. In addition, the influenza-infected bronchiectasis patients having respiratory failure were apt to develop acute kidney injury (36% vs. 16%; *p* < 0.001), and shock (47% vs. 6%; *p* < 0.001), compared to those infected with influenza without respiratory failure. A systemic steroid was administered more frequently in the influenza-infected bronchiectasis patients having respiratory failure (80% vs. 48%; *p* < 0.001). The duration of hospital stay was shorter (11.5 ± 11.4 days) among the influenza-infected bronchiectasis patients without respiratory failure than for those patients having respiratory failure (24.2 ± 15.1 days, *p* < 0.001). The overall mortality rate was 10.8%; however, the influenza-infected bronchiectasis patients having respiratory failure exhibited a significantly higher in-hospital mortality rate (27.1% vs. 6.2%; *p* < 0.001) (Table 3). As for the in-hospital mortality rate, we add a power analysis in the result. There was 73.44% power at a 0.05 level of significance. Twenty non-critical patients died of septic shock and six non-critical patients died of pneumonia with DNR consent. Because these 26 patients did not receive ventilator use, they did not fulfill the criteria for the respiratory failure group in this study.

The characteristics and outcomes of the patients with influenza confirmed by positive PCR or influenza rapid antigen test are demonstrated in the Appendix A as a sensitivity analysis. The characteristics and in-hospital outcomes of the patients with lab confirmed influenza were similar to the overall population. The influenza-infected bronchiectasis patients having respiratory failure exhibited a significantly higher one-year respiratory failure rate (17.4% vs. 5.7%; *p* = 0.012).

Influenza-associated pulmonary co-infections and their causative pathogens are listed in Table 2. 20.1% of the hospitalized patients developed a pulmonary co-infection (Table 2). The proportion of co-infections in respiratory failure patients was significantly higher compared with patients without respiratory failure (54% vs. 32%, *p* < 0.0001). The most common pathogens included *P. aeruginosa* (15%), fungus (7%), and *K. pneumoniae* (6%). *Staphylococcus aureus* (11% vs. 4%, *p* = 0.004) and fungus (19% vs. 2%, *p* < 0.001) co-infections were significantly higher in respiratory failure patients than in patients without respiratory failure.

Using multivariate analysis, age (HR 1.01, CI 95% 1.00–1.02; *p* = 0.012), previous exacerbation (HR 2.19, CI 95% 1.57–3.05; *p* < 0.001), and BACI index (HR 1.02, CI 95% 1.01–1.04; *p* = 0.015) were identified as independent factors related to hospitalization in the influenza-infected bronchiectasis patients (Table 4). The Harrell’s C-index of model (Table 4) was 0.76. Age (HR 1.04, CI 95% 1.01–1.07; *p* = 0.012), presence of shock (HR 7.66, CI 95% 3.73–15.75; *p* < 0.001) and low platelet counts (HR 0.99, CI 95% 0.99–0.99; *p* = 0.021) were independent risk factors for hospital mortality in the influenza-infected bronchiectasis patients (Table 5). The Harrell’s C-index of model (Table 5) was 0.82. The Harrell’s C-index of the significant parameters was listed in the Appendix A.

## 4. Discussion

In this study, we found that 62% patients with bronchiectasis were hospitalized after influenza infection, and 22% of the influenza-infected bronchiectasis patients developed respiratory failure during hospitalization. Patients with bronchiectasis and influenza-related hospitalization were more likely to be male, older, have more comorbidities, have worse lung function, and have had increased exacerbations in the previous year than those who were not hospitalised. Influenza-infected bronchiectasis patients with respiratory failure had a greater incidence of bacterial co-infections, acute kidney injury, shock, and higher in-hospital mortality. Age, BACI, and previous exacerbations were independent risk factors of hospitalization after influenza infection in bronchiectasis.

Viral infection is a trigger of exacerbation in respiratory diseases, but the clinical outcomes of virus infection in bronchiectasis have not been reported before. Influenza infection may cause mild to severe exacerbations in patients with bronchiectasis. Our study showed that 38% of patients with bronchiectasis and influenza infection recovered after treatment in outpatient clinic, with 11% of them acquiring a mild exacerbation after influenza infection. After influenza infection, 62% patients with bronchiectasis in our database had severe exacerbations and hospitalizations, which was compatible to previous studies demonstrating that viral infection was associated with more frequent and more severe exacerbations of pulmonary diseases [29,30]. The possible reasons might be respiratory viruses inducing airway epithelial cell sloughing and enhancing inflammation, immune cell accumulation, as well as dilatation of capillaries and parenchymal edema [31,32]. There is also an increased risk of bacterial co-infection because of impaired mucociliary clearance and parenchymal destruction in bronchiectasis [31,32]. Thus, bacterial infection is believed to be the main pathogen that causes most exacerbations of bronchiectasis, and our work provides evidence that influenza viruses also play an important role in triggering severe bronchiectasis exacerbations, possibly through bacterial co-infection. We also show that the older a patient’s age, the more BACI, and that an increased number of previous exacerbations were risk factors for hospitalization after influenza infection in bronchiectasis. Menendez et al. also found that age, severity of bronchiectasis, and more comorbidities were associated risk factors for acute exacerbations in bronchiectasis [33]. Another study also reported an increase in hospitalization among elderly men [34]. A prior history of exacerbations is demonstrated to be a predictor of future exacerbations, and patients with two or more exacerbations per year at baseline have an increased mortality risk [3]. Thus, greater bronchial and systemic inflammation in old age, greater extent of disease, and frequent acute exacerbations may contribute to perpetuating the infection-inflammation cycle and have negative impacts on prognosis in influenza infection. Our result highlights the importance of preventing or treating influenza infection in patients with bronchiectasis, particularly those with the identified risk factors. 

Severe influenza infection may progress to acute respiratory failure [15,31,35,36]. Our data indicate that 20% of hospitalized bronchiectasis patients with influenza infection developed acute respiratory failure. The rate of acute respiratory rate was higher than the previously reported 5–10% in the general population [15,32]. Primary viral infection and bacterial co-infection are the main causes of respiratory failure [15,32]. Bacterial and fungal pulmonary co-infections are associated with critical illness, such as septic shock and acute respiratory distress syndrome, leading to increased mortality rates [15,32]. Among patients with influenza-related critical illness, bacterial co-infections ranged between 10 and 30% [15,32]. Pulmonary bacterial co-infections are mainly caused by *Streptococcus pneumoniae* in Europe and *S. aureus* in the United States [12,15]. In patients with bronchiectasis, because of airway destruction and the high risk of chronic infection, the common species of bacterial and fungus co-infection after influenza may be different from those of the general population. In this study, 20.1% of the hospitalized patients developed a pulmonary co-infection, and the proportion of co-infections in the bronchiectasis patients with respiratory failure was significantly higher than in those without respiratory failure. The most common pathogens included *P. aeruginosa*, fungus, and *K. pneumoniae. S. aureus,* and fungus co-infections were significantly higher in the influenza-infected bronchiectasis patients with respiratory failure than in those patients without respiratory failure. Our previous study [19] indicated that *P. aeruginosa*, *K. pneumoniae,* and *S. aureus* were the main pathogens existing in our bronchiectasis cohort, and chronic bacterial colonization may lead to increased bacterial and decreased immune defenses, which is quickly accompanied by severe deterioration in lung function [10,37]. In animal studies, the influenza virus infection worsened the destruction of lung tissue and/or the overproduction of cytokines, increased the number of inflammatory cells in the lung, and made mice with chronic *P. aeruginosa* infection more susceptible to severe pneumonia [38]. The influenza virus is also reported to attenuate neutrophil functions *in vitro*, including the inhibition of chemotaxis, oxidative function, and lysosome secretion [39,40]. The impaired immune response following virus infection may facilitate bacterial superinfections or promote biofilm formation and subsequent disease severity or even mortality [41,42].

Seasonal influenza-related mortality is up 12% in ICU patients, especially in older adults and COPD patients [15,32]. In our population, the overall in-hospital mortality was 10% and the mortality of patients with acute respiratory failure reached 27%. Respiratory failure and multi-organ failure are the main causes of death after influenza infection [15,32]. This study showed 35% of the patients with bronchiectasis who were admitted for influenza-related infection developed acute kidney injury. Old age, shock, and low platelet counts are risk factors for influenza-related mortality. Other factors, such as comorbidities (cardiovascular, renal, liver diseases and immune deficiency) and acute kidney injury, have also been reported to be associated with increased influenza-related mortality [14,15,43]. In a pandemic influenza A study, severe infection with influenza damages the airway and alveolar epithelium, resulting in diffuse alveolar damage complicated by bacterial pneumonia, which may be another reason for the development of multiple organ failure and mortality [44]. The replication of the influenza virus in these airways and alveolar cells may contribute to the development of severe lung injury or increased susceptibility to secondary bacterial infection; therefore, limiting viral entry or replication can prevent or attenuate the severity of the infection [45,46]. Treatment with antivirus agents to stop viral replication soon after the onset of infection could improve the survival rate in patients with bronchiectasis.

Corticosteroid use is associated with increased mortality in patients with acute respiratory failure or ARDS due to influenza virus infections [15,47]. Short-term corticosteroid treatment is sometimes used to control bronchiectasis exacerbations and decrease airway inflammation [48]. Although no randomized, controlled trials could be identified to demonstrate the effects of oral corticosteroids during exacerbations, the British Thoracic Society guideline still suggests that systemic corticosteroid be used in bronchiectasis comorbid with COPD, asthma, allergic bronchopulmonary aspergillosis, or as mucoregulators [49]. In this study, more than half of the hospitalized patients received systemic corticosteroid treatment, and the usage of corticosteroids was up 80% in the influenza-infected patients with bronchiectasis and respiratory failure. Although the analysis did not reveal corticosteroids as an independent risk factor for increased mortality, corticosteroid use still should be cautiously used in patients who have bronchiectasis with influenza virus infection.

Our study has several strengths. Firstly, the CGRD database included patients from multiple medical centers and regional hospitals across Taiwan, providing strong real-world evidence of influenza infection in bronchiectasis. Moreover, the study presented 10 years of seasonal and pandemic influenza infection. To the best of our knowledge, no similar study has focused on the treatment and outcomes in influenza-associated hospitalizations in bronchiectasis to date. Secondly, we only included patients with influenza-associated infection. Previous studies included patients with bronchiectasis and viral infections from various pathogens [6,7]. This study provided a large population for investigating the impact of influenza on patients with bronchiectasis. Thirdly, we analyzed multiple outcomes of influenza infection in patients with bronchiectasis, including mild and severe exacerbation, respiratory failure and hospital mortality.

The limitations are as follows. First, we used the specific ICD-9-CM code 487 and anti-viral agents to identify episodes of influenza infection in the CGRD database. Although PCR confirmation is the gold standard of influenza infection [50], that examination is not routinely used in clinical practice in Taiwan because of the cost and time it takes to obtain the results. Besides, negative a nasal swab screen test does not rule out influenza infection [49]. However, the definition of influenza has been adopted in previous database studies and the diagnosis code of influenza infection has also been validated by comparison with laboratory data of the Taiwan CDC surveillance network [20,21,22,23]. Second, not all patients had sputum cultures performed during hospitalization. Third, we did not have complete information on the influenza vaccine in our cohort. Therefore, we could not analyze the effects of the influenza vaccine on the clinical outcomes. Fourth, treatment selection bias may exist when evaluating the effects of amantadine or systemic corticosteroids, since this was an observational study from a multi-institution database. Fifth, we acquired the data in 2018, so we only analyzed the influenza-induced exacerbations among patients with bronchiectasis before 2018. During the COVID-19 pandemic, the incidence of influenza infection might have been lower than during pre-COVID-19 periods because of public health strategy and enhanced personal hygiene. Updated data, especially after the COVID-19 pandemic, may be provided in a future study.

In conclusion, the influenza virus caused severe exacerbation in bronchiectasis, especially in those with increased age, BACI, and previous exacerbations. High risks of respiratory failure and mortality were observed in influenza-related hospitalizations in bronchiectasis. The study also demonstrated that the co-infection pathogens after influenza infection were different in bronchiectasis patients than in the general population. These results provide a better understanding of the clinical characteristics of influenza-associated infection, and the risk factors for severe exacerbation in bronchiectasis. This study highlights the importance of preventing or treating the early stage of influenza infection in bronchiectasis to lessen exacerbations and subsequent respiratory failure.

## Figures and Tables

**Figure 1 viruses-15-00537-f001:**
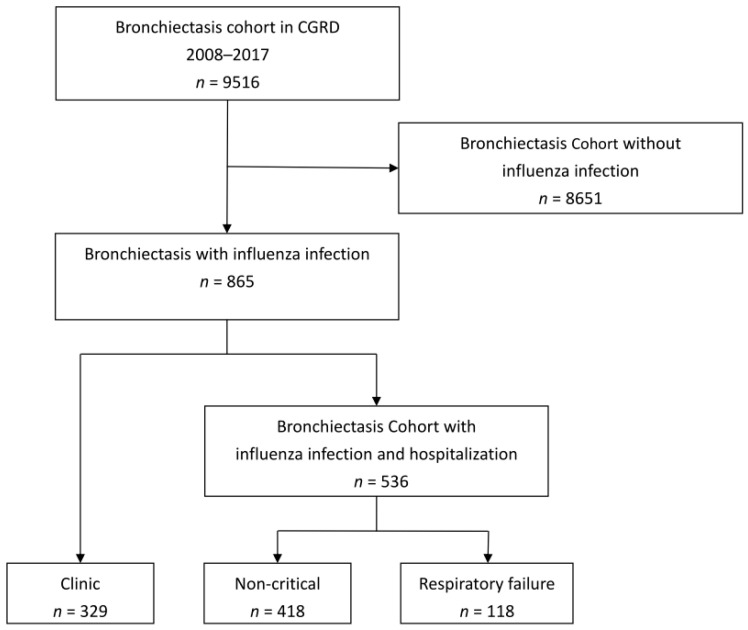
Flow diagram. CGRD: Chang Gung Research Database.

**Figure 2 viruses-15-00537-f002:**
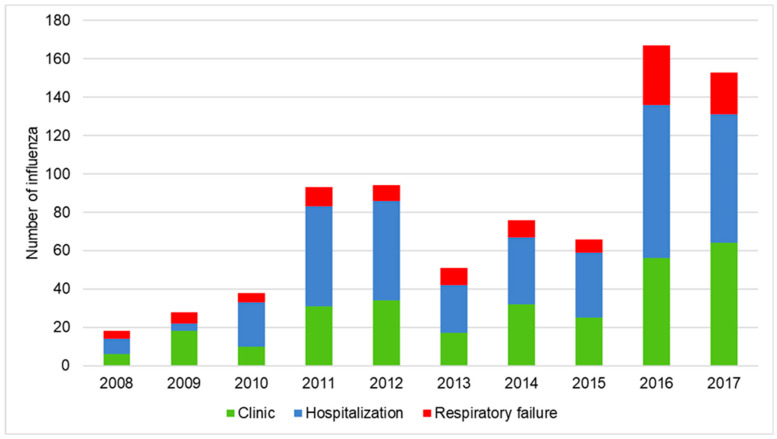
Annual incident numbers of bronchiectasis with influenza infection in CGRD since 2008–2017. CGRD: Chang Gung Research Database.

**Table 1 viruses-15-00537-t001:** Baseline characteristics of bronchiectasis patients with influenza infection.

	All *n* = 865	Clinic ^§^ *n* = 329	Hospitalization *n* = 536	*p*-Value
Age	67.9 ± 14.4	63.9 ± 14.4	70.5 ± 13.9	<0.001
Gender (Female)	447 (51.6%)	190 (57.7%)	257 (47.9%)	0.005
Etiology				
Post infection (TB)	91 (10.5%)	30 (9.1%)	61 (11.4%)	0.292
Post infection (other)	527 (60.9%)	170 (51.7%)	357 (66.6%)	<0.001
Immunodeficiencies	24 (2.7%)	9 (2.7%)	15 (2.8%)	0.956
Idiopathic	134 (15.5%)	75 (22.8%)	59 (11.0%)	<0.001
Previous exacerbation *	1.12 ± 1.8	0.5 ± 1.4	1.5 ± 1.9	<0.001
0	440 (50.9%)	250 (75.9%)	190 (35.5%)	<0.001
1	211 (24.4%)	43 (13.1%)	168 (31.3%)	
2	99 (11.5%)	20 (6.1%)	79 (14.7%)	
≥3	115 (13.3%)	16 (4.9%)	99 (18.5%)	
BACI index	7.5 ± 6.0	5.8 ± 5.4	8.6 ± 6.1	<0.001
Comorbidity				
Solid malignancy	52 (6.0%)	11 (3.3%)	41 (7.6%)	0.009
COPD	394 (45.6%)	111 (33.7%)	283 (52.8%)	<0.001
Liver disease	191 (22.1%)	73 (22.2%)	118 (22.0%)	0.952
Connective tissue disease	53 (6.1%)	12 (3.7%)	41 (7.7%)	0.017
Diabetes	231 (26.7%)	67 (20.4%)	164 (30.6%)	0.001
Chronic renal disease	141 (16.3%)	43 (13.1%)	98 (18.3%)	0.043
Cardiovascular disease	289 (33.4%)	87 (26.4%)	202 (37.7%)	0.001
Pulmonary function				0.002
FEV_1_: <50 pred.%	136 (15.7%)	40 (12.2%)	96 (17.9%)	
FEV_1_: 50–80 pred.%	188 (21.7%)	64 (19.5%)	124 (23.1%)	
FEV_1_: >80 pred.%	251 (29.0%)	119 (36.2%)	132 (24.6%)	
FVC: <80 pred.%	211 (24.4%)	86 (26.1%)	125 (23.3%)	
Influenza medicine				0.258
Amantadine	121 (13.9%)	54 (16.4%)	67 (12.5%)	
Zanamivir	121 (13.9%)	50 (15.2%)	71 (13.3%)	
Oseltamivir	614 (70.9%)	221 (67.2%)	393 (73.3%)	
Peramivir	9 (1.0%)	4 (1.2%)	5 (0.9%)	
Other Medicine				
Systemic steroid	330 (38.2%)	37 (11.3%)	293 (54.7%)	<0.001
Antibiotic	539 (62.3%)	30 (9.1%)	509 (94.9%)	<0.001

Note: BACI, bronchiectasis aetiology comorbidity index; COPD, chronic obstruction pulmonary disease; FEV_1_, forced expiratory volume in one second; FVC, forced vital capacity; pred.; predicted value; TB, tuberculosis. ^§^ The clinic group was defined as those with no episode of hospitalization. * Previous exacerbation within one year.

**Table 2 viruses-15-00537-t002:** Clinical characteristics of bronchiectasis and influenza-related hospitalization.

	All Hospitalization *n* = 536	Non-Critical *n* = 418	Respiratory Failure *n* = 118	*p*-Value
Age	70.5 ± 13.9	70.5 ± 13.5	70.4 ± 15.3	0.951
Gender (female)	257 (47.9%)	202 (48.3%)	55 (46.6%)	0.742
Etiology Post infection (TB) Post infection (other) Immunodeficiencies Idiopathic	61 (11.4%) 357 (66.6%) 15 (2.8%) 59 (11.0%)	49 (11.7%) 284 (67.9%) 12 (2.9%) 42 (10.1%)	12 (10.2%) 73 (61.9%) 3 (2.5%) 17 (14.4%)	0.639 0.216 1.000 0.182
Previous exacerbations *	1.5 ± 1.9	1.4 ± 1.8	1.8 ± 2.4	0.153
0	190 (35.5%)	154 (36.8%)	36 (30.5%)	
1	168 (31.3%)	127 (30.4%)	41 (34.8%)	
2	79 (14.7%)	63 (15.1%)	16 (13.6%)	
≥3	99 (18.5%)	74 (17.7%)	25 (21.2%)	
BACI index	8.6 ± 6.1	8.8 ± 6.2	7.9 ± 5.5	0.127
Comorbidity				
Solid malignancy	41 (7.7%)	35 (8.4%)	6 (5.1%)	0.235
COPD	283 (52.8%)	227 (54.3%)	56 (47.4%)	0.188
Liver disease	118 (22.0%)	92 (22.0%)	26 (22.0%)	0.996
Connective tissue disease	41 (7.7%)	30 (7.2%)	11 (9.3%)	0.438
Diabetes	164 (30.6%)	128 (30.6%)	36 (30.5%)	0.981
Chronic renal disease	98 (18.3%)	75 (17.9%)	23 (19.5%)	0.701
Cardiovascular disease	202 (37.7%)	167 (39.9%)	35 (29.7%)	0.041
Pulmonary function				0.101
FEV_1_: <50 pred.%	96 (17.9%)	68 (16.3%)	28 (23.7%)	
FEV_1_: 50–80 pred.%	124 (23.1%)	101 (24.2%)	23 (19.5%)	
FEV_1_: >80 pred.%	132 (24.6%)	109 (26.1%)	23 (19.5%)	
FVC: <80 pred.%	125 (23.3%)	93 (22.3%)	32 (27.1%)	
Influenza medicine				0.071
Amantadine	67 (12.5%)	48 (11.5%)	19 (16.1%)	
Zanamivir	71 (13.3%)	62 (14.8%)	9 (7.6%)	
Oseltamivir	393 (73.3%)	305 (72.9%)	88 (74.6%)	
Peramivir	5 (0.9%)	3 (0.7%)	2 (1.7%)	

Note: BACI, bronchiectasis aetiology comorbidity index; COPD, chronic obstruction pulmonary disease; FEV_1_, forced expiratory volume in one second; FVC, forced vital capacity; pred.; predicted value; TB, tuberculosis. * Previous exacerbation within one year.

**Table 3 viruses-15-00537-t003:** Clinical parameters and main clinical outcomes of hospitalization.

	All Hospitalization	Non-Critical	Respiratory Failure	
	*n* = 536	*n* = 418	*n* = 118	*p*-Value
Sputum culture	384 (71.6%)	281 (67.2%)	103 (87.3%)	<0.001
Positive sputum culture	147 (38.3%)	91 (32.4%)	56 (54.4%)	<0.001
*Pseudomonas aeruginosa*	57 (14.8%)	38 (13.5%)	19 (18.5%)	0.229
*Klebsiella pneumoniae*	24 (6.3%)	16 (5.7%)	8 (7.8%)	0.457
*Haemophilus influenzae*	20 (5.2%)	17 (6.1%)	3 (2.9%)	0.220
Fungus	27 (7.0%)	7 (2.5%)	20 (19.4%)	<0.001
NTM	11 (2.9%)	7 (2.5%)	4 (3.9%)	0.494
*Staphylococcus aureus*	23 (5.9%)	11 (3.9%)	12 (11.7%)	0.005
*MDR-AB*	11 (2.9%)	6 (2.1%)	5 (4.9%)	0.173
Laboratory data				
WBC (×10^3^/uL)	10.5 ± 5.5	10.1 ± 4.9	12.1 ± 6.7	0.003
Platelet (×10^3^/uL)	208.8 ± 94.5	208.3 ± 91.3	210.8 ± 105.4	0.814
C-reactive protein (mg/L)	92.6 ± 93.1	84.2 ± 85.5	117.1 ± 109.0	0.005
Creatinine, baseline (mg/dL)	1.3 ± 1.6	1.2 ± 1.4	1.6 ± 2.0	0.084
Creatinine, ward (mg/dL)	1.5 ± 1.8	1.3 ± 1.6	2.0 ± 2.3	0.002
Acute kidney injury	108 (20.2%)	66 (15.8%)	42 (35.6%)	<0.001
Influenza medicine				0.071
Amantadine	67 (12.5%)	48 (11.5%)	19 (16.1%)	
Zanamivir	71 (13.3%)	62 (14.8%)	9 (7.6%)	
Oseltamivir	393 (73.3%)	305 (73.0%)	88 (74.6%)	
Peramivir	5 (0.9%)	3 (0.7%)	2 (1.7%)	
Other medication				
Systemic Steroid	293 (54.7%)	199 (47.6%)	94 (79.7%)	<0.001
Antibiotic	509 (95.0%)	394 (94.3%)	115 (97.5%)	0.161
Inhalation gentamicin	27 (5.0%)	19 (4.6%)	8 (6.8%)	0.327
Days of hospitalization	14.3 ± 13.3	11.5 (11.4)	24.2 ± 15.1	<0.001
Shock	83 (15.5%)	27 (6.5%)	56 (47.5%)	<0.001
In-hospital mortality	58 (10.8%)	26 (6.2%)	32 (27.1%)	<0.001

Note: MDR-AB, multidrug-resistant Acinetobacter baumannii; NTM, non-tuberculosis mycobacteria; WBC, white blood cell count.

**Table 4 viruses-15-00537-t004:** Univariate and multivariate analysis of influenza-related hospitalization.

	Univariate Analysis			Multivariate Analysis		
	HR	95%CI	*p*-Value	HR	95%CI	*p*-Value
Age	1.02	1.01–1.02	<0.001	1.01	1.00–1.02	0.012
Female	0.84	0.71–0.99	0.038	0.86	0.69–1.07	0.178
Etiology						
Post infection	1.39	1.16–1.66	0.001	0.91	0.69–1.18	0.472
Idiopathic	0.61	0.47–0.80	0.001	0.67	0.39–1.12	0.128
Previous exacerbations *						
0	Ref.			Ref.		
1	2.27	1.84–2.79	<0.0001	2.12	1.60–2.79	<0.001
2	2.30	1.77–3.00	<0.0001	2.03	1.42–2.91	0.001
≥3	2.54	1.98–3.25	<0.0001	2.19	1.57–3.05	<0.001
BACI index	1.04	1.02–1.05	<0.0001	1.02	1.01–1.04	0.015
Pulmonary function						
FEV_1_: <50 pred.%	1.63	1.16–2.28	0.005	1.23	0.87–1.74	0.235
FEV_1_: 50–80 pred.%	1.46	1.07–1.99	0.017	1.26	0.92–1.72	0.150
FEV_1_: >80 pred.%	Ref.			Ref.		
FVC: <80 pred.%	1.22	0.94–1.58	0.142	1.07	0.82–1.39	0.614

Note: BACI, bronchiectasis aetiology comorbidity index; FEV_1_, forced expiratory volume in one second, FVC, forced vital capacity. * Previous exacerbation within one year.

**Table 5 viruses-15-00537-t005:** Univariate and multivariate analysis of in-hospital mortality.

	Univariate Analysis			Multivariate Analysis		
	HR	95%CI	*p*-Value	HR	95%CI	*p*-Value
Age	1.04	1.01–1.06	0.002	1.04	1.01–1.07	0.012
Female	0.55	0.32–0.95	0.032	0.62	0.35–1.11	0.111
Etiology						
Tuberculosis	2.43	1.30–4.51	0.005	1.94	0.99–3.81	0.053
Cardiovascular disease	0.48	0.26–0.87	0.016	0.69	0.36–1.34	0.275
Platelet count	0.99	0.99–0.99	0.015	0.99	0.99–0.99	0.021
Acute kidney injury	3.76	2.22–6.35	<0.001	1.63	0.91–2.91	0.102
Systemic steroid	1.85	0.99–3.46	0.053	0.71	0.35–1.46	0.352
Respiratory failure	2.00	1.16–3.45	0.012	1.45	0.78–2.69	0.239
Shock	7.68	4.36–13.55	<0.001	7.66	3.73–15.75	<0.001

## Data Availability

The data are not publicly available due to ethical restrictions and regulations of the Institutional Review Board of Chang Gung Memorial Hospital.

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
