# Peer review of "Risk Factors for Influenza-Induced Exacerbations and Mortality in Non-Cystic Fibrosis Bronchiectasis"

_viruses, 2023, doi:10.3390/v15020537_

Round 1

Reviewer 1 Report

Dear Editor and Esteemed Authors,

I had the pleasure to review this work which you so kindly asked me to do so titled “Risk factors for influenza-induced exacerbations and mortality in non-cystic bronchiectasis” by Dr. Huang and his colleagues from Taiwan.

This is a multi-institution, retrospective analysis using database mined data and variables which tries to identify the effect of influenza infections on patients with bronchiectasis. It utilized data from 865 patients of which 536 needed hospitalization. The analysis concluded that patients with influenza-related hospitalization with bronchiectasis had a higher mortality and incidence of respiratory failure. The authors were also able to demonstrate factor associated with this.

This is a well conducted study even if it is database based. It utilizes a clearly established methodology and variables set with pre-established definitions. The analysis is adequate although as I mention below it could be improved. The manuscript is well written and easy to understand with only a few English mistakes, nothing a bit of polishing up by a native speaker or a professional editing service can’t fix. The tables and graphs included are easy to read and understand.

Overall, this is a nice paper so I only have some very minor comments:

Comments:

1.       Why was not a sample size or power analysis performed prior to the collection of data/variables and enrollment of patients to demonstrate the sample size needed for statistically meaningful results? This is easy to do at the beginning stages of the study and adds a lot of value to the validity of the outcomes reported. 

2.       Why was not an ROC analysis performed to validate the COX model? 

3.       26/6.2% of non critical patients died while hospitalized. Why is that? What were the causes of mortality if not respiratory failure?

In conclusion, I think this a good little study with a somewhat interest hypothesis/aim which although one would think is well known and established has limited available good evidence. Therefore, I think this work should be presented to the scientific community following some minor edits/corrections. Kind regards to all.

Author Response

  1. Why was not a sample size or power analysis performed prior to the collection of data/variables and enrollment of patients to demonstrate the sample size needed for statistically meaningful results? This is easy to do at the beginning stages of the study and adds a lot of value to the validity of the outcomes reported.

Response:

We appreciate the valuable comments. This was a retrospective study based on CGRD database, so we did not calculate the sample size. As for the in-hospital mortality rate, we add a power analysis in the result. There was 73.44% power at a 0.05 level of significance. We have mentioned the point in the revised manuscript on page 6, lines 10-12.

  1. Why was not an ROC analysis performed to validate the COX model?

Response:

We appreciate the valuable suggestion. We add an ROC analysis to validate the COX model in the result. The Harrell’s C-index of model (Table 4) was 0.76. The Harrell’s C-index of model (Table 5) was 0.82. The Harrell’s C-index of the significant parameters were listed in the supplement. We have addressed the results on page 7.

  1. 26/6.2% of non critical patients died while hospitalized. Why is that? What were the causes of mortality if not respiratory failure?

Response:

20 non-critical patients died of septic shock and 6 non critical patients died of pneumonia with DNR consent. Because these 26 patients did not receive ventilator use, they did not fulfil the criteria of respiratory failure group in this study. We addressed this information in page 6, lines 12-15.

Reviewer 2 Report

This study investigated the clinical manifestations of influenza-induced exacerbation among patients wtih bronchiectasis. Ovearll, the study is interesting and the manuscript is well-written. I just have comments.

1. Please explained why upto 50% of patients received systemic corticosteroid in this study. Is it possible for COPD with exacerbation rather than bronchiectasis

2. I have serious concern about the used definition of co-bacterial infection based on culture results. How do you differentiate colonization or infection?

3. The study period ended with 2017, which was six years ago. A updated data, especailly after COVID-19 pandemic could be added.

Author Response

  1. Please explained why up to 50% of patients received systemic corticosteroid in this study. Is it possible for COPD with exacerbation rather than bronchiectasis

Response:

Short-term of corticosteroid treatment is sometimes used to control bronchiectasis exacerbations and decrease airway inflammation. Although no randomized controlled trials could be identified to demonstrate the effects of oral corticosteroids during exacerbations, the British Thoracic Society guideline still suggests systemic corticosteroids to be used in bronchiectasis comorbid with COPD, asthma, allergic bronchopulmonary aspergillosis or used as mucoregulators [1]. We have added this point in the Discussion section, page 10, lines 16-20.

Reference:

[1]    Adam T Hill, Anita L Sullivan, James D Chalmers, Anthony De Soyza, J Stuart Elborn, R Andres Floto et al: British Thoracic Society Guideline for bronchiectasis in adults. Thorax. 2019, 74(Suppl 1), 1–69.

  1. I have serious concern about the used definition of co-bacterial infection based on culture results. How do you differentiate colonization or infection?

Response:

Thank you for the valuable comment. The definition of co-bacterial infection was based on the presence of pneumonia and positive culture results. We modified the definition in the Method as “ Pulmonary bacterial co-infection was defined as the presence of pneumonia and a positive bacterial culture in sputum or bronchoalveolar lavage fluid during the period of hospitalization.” in page 3.

  1. The study period ended with 2017, which was six years ago. A updated data, especailly after COVID-19 pandemic could be added.

Response:

This is a multi-institution, retrospective analysis using CGRD database. We acquired the data in 2018 so we only analyzed the influenza-induced exacerbations among patients with bronchiectasis before 2018. During COVID-19 pandemic, the incidence of influenza infection could be lower than pre- COVID-19 periods because of public health strategy and enhanced personal hygiene. An updated data, especially after COVID-19 pandemic could be conducted in future study. We have addressed this suggestion in the section of limitations, the bottom of page 10.

Round 2

Reviewer 2 Report

the authors response well, so I have no more comment.